# Impact of Coaching on the Development of Personal and Social Competences among Secondary School Students

**DOI:** 10.3390/children10061025

**Published:** 2023-06-07

**Authors:** Andrés Fernando Avilés Dávila, María Eugenia Martín Palacio, Cristina Di Giusto Valle

**Affiliations:** 1Department of Research and Psychology in Education, Complutense University of Madrid, 28040 Madrid, Spain; mariaeugeniamartin@edu.ucm.es; 2Department of Educational Sciences, University of Burgos, 09001 Burgos, Spain; cdi@ubu.es

**Keywords:** coaching, competences, effective personality, secondary school, dimensions, students

## Abstract

In this study, the impact of school coaching in Spain on the dimensions that comprise the effective personality construct and the development of those dimensions among secondary school students is analyzed. Differences relating to the variables of gender, course year/age, and the state/private sector of each educational center are specified. A longitudinal study employing a quasi-experimental methodology was conducted with a sample of 310 students in attendance at 6 educational centers within the Community of Madrid, Spain; the study cohort comprised 156 (50.3%) male students and 154 (49.7%) female students. The effective personality questionnaire in secondary education was used for the pre- and post-intervention evaluations, as well as to conduct an open survey once the program had ended. The students who attended the six-session intervention expressed high degrees of satisfaction. Despite the brevity of each session (45–60 min), a slight non-significant improvement was observed in the experimental group relative to the control group in the dimensions that were studied (*p* > 0.05). Regarding gender, the program worked better with male rather than with female students. It was also more effective within state-aided than in state schools.

## 1. Introduction

Nowadays, we find ourselves in a very diverse and interconnected society where people are not only required to possess technical-scientific training, but also to hold inter- and intra-personal skills with which they can develop their personal and school relationships and maintain them within globalized environments [1,2].

Factors relating to successful personalities and behavioral capabilities under certain circumstances with different levels of difficulty have been explored in various studies. It can be understood as a competition between different dimensions of the personality [3]. Researchers have considered the development of personal and social competencies through the effective personality construct (EPC). The EPC is based on the premise that the acts or the behavior of an effective personality are intelligent and have trainable behavioral competences [4]. A set of competencies is grouped together in the EPC that has been separately studied through such theories as Bandura’s theory of self-efficacy [5,6,7], Gardner’s multiple intelligences [8], Sternberg’s triarchical intelligence theory [9], Goleman’s emotional intelligence [10], Salovey, Woolery and Mayer’s measurement tools [11], Heath’s conceptions of maturity [12], and Zacares and Serra’s work on personal maturity [13], studies on self-conceptualization, motivation, expectations, and the bar-on model for problem solving and social skills [14], and Garmezy and Masten’s work on competence indicators [15]. A particular structure is proposed in the study of Martin Del Buey et al. [16] based on the above-mentioned theories.

As a consequence of this process, the effective personality is formed by 10 competencies around a person. These competencies are grouped into four categories [3]. The first category, “Strengths”, encompasses the dimensions of self-conceptualization and self-esteem. The second category, “Demands”, groups together the dimensions of motivation, attribution, and expectations. The third category, “Challenges”, constitutes the dimensions of facing up to problems and decision making. Finally, “Relationships”, the fourth category, is focused on the dimensions of empathy, assertiveness, and communication.

Starting with this structure [16], an effective person can be defined in the following terms:

People with effective personalities have self-knowledge and self-esteem (self-conceptualization and self-esteem) and are constantly maturing (at any stage of their development). They have the capability (intelligence) to achieve (efficiency), controlling the causes (attribution of causality) of any consequences (success or failure). They confront personal, circumstantial, and social difficulties (facing the problem) that arise, taking suitable decisions without jeopardizing their good relationships with others (empathy and communication); nor do they jeopardize these relationships in the service of purely personal aspirations (assertiveness) [p. 35].

Over the past two decades, coaching has rapidly moved from the world of sports into business and social arenas, where coaching is, for example, provided to victims of domestic violence, to clients of social services, in the family, and, above all, in education to students and teachers [17]. Coaching is therefore perceived as a procedure that favors the effective professional development of a person, not only from a technical, but also from an emotional point of view, in so far as people are attuned to their own way of life [18,19,20,21].

We can consider coaching in the educational context as a process of assisting with the achievement of preset objectives. It encourages students to think for themselves and to be responsible for their own learning. Coaching can therefore reinforce and maintain an effective course of action that enables individual to sustain and strengthen their competences beyond school. The Socratic method is primarily used, favoring an inward perspective to help people to discover their full potential, of which they may as yet be unaware. Following that approach, a wide variety of complex and diverse challenges may be explored in everyday activities [22,23,24,25,26].

The coaching approach fits very well with the constructivist educational model as it is supported by the action of the learner and self-discovery. When pedagogy is conceived as a practice of freedom and autonomy, coaching creates a scenario that contributes favorably to teaching and learning processes [25,27].

The conversation we are interested in creating with students through coaching is one that generates security, trust, and mutual understanding. It will not only build a good working environment, but it will also provide participants with a space for reflection and for listening to experiences. It should be place where important and complex issues can be mulled over in a way that leads to action, contributing to the design of new alternative learning models. In this scenario, the teacher sheds the role of director to become a facilitator of student learning. In other words, the teacher-coach helps students to go “where they want to go”, bringing out their full potential [28,29,30,31].

Having said as much, school coaching can also develop several competencies based on a search for excellence rather than deontic requirements, an approach which can also yield satisfactory academic results.

Given the above factors, the objective of this research is, from an empirical point of view, to estimate the influence of the coaching process on the development of personal and social competencies. These competencies, both general and specific, are based on the EPC. Whether this tool favors the development of these competencies when used with secondary school students is also corroborated in this study, and differences depending on gender, course year/age, and the type of educational center (state-aided/state schools) are also verified.

## 2. Materials and Methods

In this research, a quasi-experimental methodology was used with 8 experimental groups and 8 control groups. A longitudinal design was carried out by evaluating the students at two different times, with a time interval of one-and-a-half to two months each (pre-test and post-test). In this study, the two measures of the dependent variables were taken from different groups: the groups that received the intervention and other similar groups, not created at random, that did not participate in this coaching process [32]. This type of group is called “non-equivalent” as it is not randomly formed [33].

The independent variable of the study was the intervention in school coaching, while the dependent variables were the personal and social competences described in the EPC.

### 2.1. Sample

The participants in this research were selected using non-probability convenience sampling. The sample comprised 310 students attending Spanish Secondary Schools from six centres (four state and two state-aided schools) of the Autonomous Community of Madrid (Spain). The gender distribution was 156 (50.3%) males and 154 (49.7%) females between the ages of 12 and 18. This sample was divided into a control group of 154 (49.7%) students and an experimental group of 156 (50.3%).

The convenience sample was formed of contacts at educational centers within the Community of Madrid who had been invited to participate in a study on “Effective Personality and School Coaching”. Since the students were minors, the provisions of Section IV (art. 34) of the Deontological Code of Psychology [34], in which the informed consent of their family is mentioned, were carefully observed.

### 2.2. Instrument

A questionnaire with the following open-ended questions was administered to the students to ascertain their level of satisfaction with the program: How did we function as a group? What did you learn from this experience? What was useful for you during the intervention? How did you feel during the process? What did you dislike about the experience? Additionally, what was your final conclusion?

The questionnaire on effective personality in secondary education (CPE-Es/Ch), prepared by Dapelo et al. [35], was used for the quantitative study related to the possible effectiveness of the change. The instrument consists of 23 items and has a Cronbach’s alpha reliability of 0.83 in relation to the Spanish population [36]. The response scale was on a Likert-type scale with five alternatives. It was used to evaluate the four areas of effective personality through four factors: the self-esteem factor (4 items), the academic self-realization factor (8 items), the resolutive self-efficacy factor (5 items), and the social self-realization factor (6 items).

### 2.3. Procedure

With each of the schools, a schedule of activities was established according to their availability and requirements. At these meetings, it was also determined which groups would participate in the intervention. The study was conducted between February and May 2016. Before completing the questionnaires and starting the coaching process, the parents were asked for their authorization and the object of the investigation was explained to them.

Six 45-to-50-min sessions were scheduled on a weekly basis for the development of each experimental group. Each coaching intervention had been developed by the researcher, who was responsible for building trust, encouraging the expression of ideas and emotions, and stimulating individual and collective work, as well as respect, tolerance, and acceptance. A guide was developed for this intervention, (Table 1) where the activities to be carried out during the whole process of school coaching were mentioned.

At the end of each coaching process, information was collected through an open survey related to the participant’s perception of the process. These data will be taken into account in future interventions.

The administration of the effective personality questionnaire in the pre-test phase (both control and experimental group) was carried out in a single collective session per classroom during academic hours. The investigator performed this task and firstly explained the objectives of the research to both give transparency to the process and to diminish any possible resistance to the administration of the questionnaire. The students were told to be as sincere as possible, that there were no correct or incorrect answers, and that the confidentiality of their data was guaranteed at all times. In the event of an incident of any sort, the students were individually assisted. It is worth mentioning that no—economic or material—remuneration was offered to the students who participated in this research. At the end of the intervention period, the (post-test) questionnaire was issued once again under the same conditions.

### 2.4. Data Analysis

Once the different options had been classified, the corresponding percentages were obtained for each open-ended question in the survey for the qualitative analysis of the responses.

The Kolmogorov–Smirnov test establishes whether a data set has a normal distribution for a quantitative analysis. Depending on the result, the difference analyses must be performed with parametric tests (if they present homoscedasticity) or non-parametric tests (if their distribution does not follow the normal curve). The differences in the pre-test between the control group and the experimental group were studied to check the homogeneity of the starting point of both groups, using the Student’s *t* test for the independent samples with a normal distribution and the Mann–Whitney U test for the non-parametric samples.

A study of differences between the pre-test and post-test results was conducted in both groups to gauge the effectiveness of the program. W used with the Student’s *t* test and the Wilcoxon test for the dependent samples with a normal data distribution and for the non-parametric data sets, respectively. Finally, a study of differences in the post-test between the control and the experimental group was conducted in order to check for any changes at the end of the study. Once again, we used the Student´s *t* test for independent samples with parametric distributions and the Mann–Whitney U test for the non-parametric ones. In addition, an ANCOVA test of the experimental group post-test results (comparing any differences with the pre-test) was performed to verify whether there were significant differences in the intervention, both in relation to the general EPC scale and in terms of the variables gender, course year/age, and type of educational center (state-aided/state schools). All these analyses were performed with the statistical program SPSS 20.0.

## 3. Results

The results of the open survey are presented below in various graphs.

The responses to the question ‘How did we function as a group?’ are shown in Figure 1. Slightly over 76% of the students clearly expressed a good or very good level of satisfaction with the functioning of the group.

The responses to the question ‘What you have learned from this experience? Has it given you greater self-knowledge, knowledge of your peers, and awareness that respect is important in relationships with other people?’ are shown below in Figure 2.

The responses to the question ‘What was useful for you during the intervention?’ are shown below in Figure 3. The appraisals varied greatly, the most prominent of which was related to the materials in use. This was followed by 29.49% of students who mentioned that the activity had helped them to value themselves properly, giving them an opportunity for reflection and group activities.

The responses to the question ‘How did you feel during the process?’ are shown below in Figure 4. A significant majority of students (91.03%) during this experience felt satisfied and optimistic as they had the perception of being listened to and respected among their peers.

The responses to the question ‘What did you not like about the experience? are shown below in Figure 5. The majority of students (48.08%) considered that the experience had been a beneficial one for them, with more critical attitudes being expressed towards reflection, the lack of collaboration, and the short duration of the sessions.

The responses to the final question ‘What was your conclusion?’ are shown below in Figure 6. The perception of a significant number of the students was that their self-knowledge had increased after the program ended.

The descriptives of the global scale and its areas are presented for both the pre-test and the post-test of the control and experimental groups (Table 2) in order to rate the effectiveness of the program. It was observed that the averages in the control group decreased, which occurred both on the global scale and in each area. They increased on a global scale and in all areas for the experimental group, except for challenges, where they decreased.

An ANCOVA test was applied to check whether the improvement or worsening of the post-test results against those of the pre-test was due to the effectiveness of the program. This parametric test was chosen as the size of both groups was similar and there were over 30 participants in each [37,38,39]. The results in Table 3 showed that the students participating in the intervention had neither significantly improved nor worsened with regard to the control group, either on the global scale or in their areas (*p* > 0.05). It was also observed that the effect size was low (*η*^2^*_p_* > 0.20), which corroborates the non-existence of differences. Nevertheless, it is worth mentioning that the global scale of effective personality indicated that there was a significant difference in a marginal nature (*p* < 0.100), underling a trend toward effective school coaching interventions at a general level.

The ANCOVA test was applied to the results of the experimental group—controlling for differences with the pre-test—in order to know whether there were significant differences in the intervention according to the variables of gender, academic year/age, and the type of educational center (state-aided/state schools). The results pointed to significant gender differences in the area of challenges (Table 4), in which men improved significantly more than women (whose post-test scores were on occasions even lower compared to the pre-test results) (*p* = 0.032). In addition, a slight trend in the area of strengths suggested that the program functioned better among the groups of men (*p* = 0.088). As for the general scale and the areas of demand and relationships, there was no differential functioning of the program (*p* > 0.05) concerning gender.

On the other hand, the course year/age variable had no bearing on the perception of how well the program functioned (*p* > 0.05), either on a global scale or in any of the areas. With respect to the type of educational center (state-aided/state schools), there was a slight tendency in the area of relationships that led us to suggest that the intervention could have functioned better in state-aided schools (*p* = 0.074). Furthermore, the intervention was perceived to function in the same way on a general scale as it did in other areas (*p* > 0.05). The three variables showed a low effect size (*η*^2^*_p_* > 0.20).

## 4. Discussion

The purpose behind this research was to estimate, from an empirical standpoint, the influence of group coaching on the development of both the general and the specific personal and social competences defined in the EPC. It was also to corroborate whether using this tool in Spain with secondary school students favors the development of EPC competences and whether there were differences depending on the variables of gender, course year/age and the type (state-aided/state schools) of educational center.

The first point to keep in mind is that the intervention involved secondary school students. These adolescents are undergoing normal developmental changes that affect both physical and psychological–cognitive–social aspects of life, including the development of identity, which is characterized by systematic growth toward maturity and a search for stability [40,41,42,43,44]. In the opinion of Rother et al. [42], the new generations of adolescents are in no way similar to the preceding generations. Today, the desire of many teenagers is to live out their lives very actively and with some impatience to enjoy the ‘here and now’ to the fullest. These 14–18-year-old teenagers are in a stage of exploration [45] where they are still developing their consciousness along with rapid and intense changes that shape their natural development. Both parents and the socio-educational environment can play an important role in avoiding what is known as adolescent egocentricism, thereby favoring wellbeing and the development of effective social relations [46,47,48,49], as adolescents nowadays grow up in a social environment that is all too often hedonistic, mediocre, and vulgar [50].

On the one hand, it is important to point out that each time the students exercise a certain level of competence, they can develop it within themselves, though sufficient time is needed to reflect on dilemmas and alternative solutions in order to gain greater maturity with competence and to integrate it into their behavior. On the other hand, according to García San Pedro [51], competencies involve arduous performance and are obtained through the shared work of several subjects over lengthy periods of time. Additionally, teachers agree in that their tutorial activities are focused on activities related to the organization of the events of each school year, such as preparing street markets, organizing the cultural week, sports, museum visits, and other extracurricular activities, solving problems with teachers and, in general, seeking solutions to disagreements or divergences that may arise in each group and that are consequently quite alien to the development of the personal and social skills analyzed in this study. In contrast, Batista et al. [52,53,54] commented that these types of extracurricular activities favor student wellbeing. Thus, one key teaching role is to provide students with abundant opportunities to experience small victories and to celebrate small successes in academia and beyond the confines of the classroom [55,56,57], thereby fostering self-efficacy and the potential it has in the classroom.

According to research carried out by Almeida et al. [1], emotional intelligence and emotional education can be worked on and improved over time, with very favorable results for the person. Over the last 4 years, the two state-aided schools under study have integrated activities surrounding emotional education into the first years of schooling. Teachers receive suitable training for the development of the activity, although there is no continuity in this regard secondary school education, regardless of any other teacher training and involvement in this type of work. The development of emotional competences is not integrated into the educational activities of any training cycle in schools within the public sector. It is a type of training that is specifically limited to some extracurricular activities that are in the background. Hence, both the teachers and the educational center in question must prioritize the educational itinerary.

The results must be assessed in that context. School coaching was used during the intervention. There have been similar studies with favorable results within other contexts, in which the intervention has been complemented with training programs on areas such as emotional intelligence and multiple intelligence, conflict management, and communication [58,59,60,61,62,63,64]. Furthermore, according to some authors, it appears that academic coaching is related with a student’s academic persistence, skills acquisition, and development, although the conclusions of some other investigations are that there is only a slight improvement in the grade point average of students after the intervention [65,66,67].

From a qualitative point of view, the results of the open survey administered to students at the end of the program reflected high satisfaction levels with personal issues, the functioning of the group, and the group activities. Additionally, they evidence greater self-knowledge, and self-esteem (one of the appraisals that acquired greater value), capacity for reflection, assessment of group work, the value of perseverance, and consideration of perspectives other than one’s own. The results were in line with the work of Canaan et al. [68] on the effectiveness of this type of process at increasing student achievement. These authors for more investigations of this sort with focused sessions over time. It is important at this stage to review the influence of the group and the evaluations of the experimental group members participating in the program. According to the studies of Sutton [69], a coaching process in a group setting not only increases the self-awareness and the commitment of the participants, but it also provides a supportive environment for the expression of emotions.

It is important to consider that these assessments are in line with the development of personal and social skills that postulate the construction of an effective personality where self-esteem, self-realization, social relations, and resolution capacity play important roles in development.

The following discussion concerns the questionnaire scores. Firstly, the global scores indicative of the development of personal and social competences in both the experimental group and the control group were low since, out of a maximum possible score of 115 points, the averages never exceeded 86 points, although it may be noted that they were all slightly over the average level. The most significant weaknesses were presented in a specific order and each competence was analyzed in terms of resolutive self-efficacy followed by academic self-realization, self-esteem, and social self-fulfillment. According to Ackerman [64], decision making constitutes a learning process that continuously occurs throughout life and that requires adaptation to new personal and social contexts.

Secondly, it is important to point out that although the differences obtained in the pre-test and post-test were not significant among the students who received the school group coaching intervention in its four areas (*p* > 0.05), it is worth observing that there was a significant difference of a marginal nature in the global scale of effective personality (*p* < 0.100). It points to a trend towards effective interventions for group coaching in school at a general level. Given the brevity of the intervention, it is also clear that higher differential indices could hardly be expected. However, it is necessary to add some considerations that were collected from the teachers on the possible effect of the educational group coaching intervention the experimental group. Among the six teachers who were interviewed, two shared the view that most students had expressed interest and appeared to have been involved in each session, especially those students who might not otherwise have shown interest in a normal class. Two others mentioned that the methodology had allowed them to focus on the work that was conducted. Three teachers had the perception that the intervention had led them to reflect on important situations that they had not stopped to think about before; to look at themselves; to open themselves up a little more; and to express what was happening to them. In the end, they affirmed that they had learned more and realized that there were other ways of acting and achieving goals. Four teachers had the perception that the most important change had been in the improvement of the interpersonal relations, not only between students, but towards the teacher, with greater participation on the part of the group in the classes, improving their state of emotional wellbeing [69]. An important majority of teachers agreed that there were critical factors that could influence the normal development of this activity: for example, if the teachers considered the classes to be difficult groups for work to advance through properly because of disruptive behaviors that obstructed normal classroom routines and particular activities, or the fact that the intervention took place in the third quarter of the school year.

Other considerations must also be added. This is due perhaps to students overvaluing and/or overestimating their own potential capabilities in their responses to the pre-test survey. According to Tursunova [49], the type of reflection that teenagers make is similar to a model of egocentrism known as a personal fable, which consists of overvaluing their characteristics. Therefore, in adolescence, an important challenge is posed. This is the development of a coherent sense of identity, a process which allows the adolescent to make the most acceptable decisions with regard to future situations [44,70]. Another consideration is that once the intervention has begun, students find themselves in a situation where participation is dynamic and inclusive. As the intervention process progresses, it becomes necessary to incorporate a series of aspects, such as knowing how to be (responsibility, respect, involvement, and commitment) and knowing how to do (application of knowledge). When living out the experience in reality, the students may have performed a none-too-critical or -constructive analysis of each experience, which might have distracted them in the process of assimilation and action, with consequences for one or several of the dimensions that were studied. The above comment is in line with the view expressed in [51]: the commitment must be presented in a “continuum”. In other words, it must be presented gradually in the different stages of the pedagogical process, and it will in that way be closely linked to an ability to increase student commitment.

The results regarding gender indicated that the program worked better with men in the area of resolution of self-efficacy and self-esteem, and better with women in the area of strengths. This may have been due to the fact that women at this developmental stage tend to feel more reserved and insecure when facing some difficulty with the resources they possess. As Rosario Alba [71] stated, male adolescents clearly and unequivocally employed important levels of autonomy, which is expressed in their daily work with respect to women. However, research conducted by Carrion et al. [72,73] showed that women adjusted to school better because they were willing to adapt and had little perception of academic frustration, while men were more indifferent and felt greater pressure to succeed in their studies. In addition, a range of studies [74,75,76] have reported high ratings in both social competencies and self-efficacy among women. Furthermore, some authors stated that women handled emotions better [77], while others have mentioned that men possess greater emotional intelligence [78].

In relation to the course year/age variable, it was observed that the average results of the second-year students increased on a global scale and in the post-test; among the third-year students, the results decreased on a global scale and in all areas, except for strengths, where the post-test scores were a little higher than the pre-test scores; and in the fourth year of the course all the scores increased, except for the area of challenges, where the post-test results were a little lower than the pre-test results, meaning that the program was seen to function in different ways in none of the areas. This may be explained by the different developmental stages of maturity during this stage among the students. Rosario Alba [71] stated that, as the students walk through their adolescent years, on the one hand, they leave aside changes and may be able to provoke a series of modifications in their personal environment; on the other hand, they undergo personal changes that are convenient to keep in mind as greater maturity leads on to adulthood. Some people tend to mature more quickly than others at any age because the process of maturity and the seeds of a mature personality are sown early on in life [79]. With regard to the impact of this intervention in relation to the (state-aided/state school) type of educational center, there was a slight tendency within the state-aided centers towards better functioning of the program in the area of social self-realization. This may be due to the fact that the formative cycle of infant education in these educational institutions has included activities in their learning paths related to emotional intelligence. There is a greater impetus to develop extracurricular activities that have to do with the development of intrapersonal and interpersonal competences at higher grades. In the opinion of [80], the culture of the school should be one of a space where seeds are planted to enrich the teaching and learning of both students and teachers.

As stated by Hunt [81], it is important for the collaboration of all those agents who can intervene in the integral development of the student to develop a participatory vision of teaching. This was a factor that could not, unfortunately, be explored in this research. The education of children, youth, and adolescents has to be a commitment of the whole society in general. School should not therefore be perceived as a place where one acquires all the learning that a person needs to integrate into a social grouping. In this way, as Castillo-Ceballos [82] stated: “The society that educates assumes responsibility for the ongoing formation of all its members. To that end, it promotes spaces for coexistence and social participation” [2016, p. 71]. In addition to the above, it should be noted that in order to promote the development of autonomy and independence among adolescents, the family, in many cases, needs to decrease the level of control, provide support, understand feelings, and incorporate limits as the adolescent moves towards adulthood [83].

Similarly, the number of school coaching sessions—six—has been pointed out, as has the time dedicated to each one of them—between 45 and 50 min. These factors are fundamental, as Boniwell et al. [84] mentioned, because knowing the temporality of each session will intensify the wellbeing and productivity of the beneficiaries which, in this case, is the students. This point is for practical purposes contemplated in the first session, where when defining the contract between both parties—coach and coachee—sufficient information is given on process durability, the sessions, and the way in which techniques can be implemented.

## 5. Limitations and Future Implications

It is worth mentioning that some of the limitations encountered include the difficulty of subsequent monitoring due to the circumstances and demands of each school. This explains why there are no references to say that these changes have been maintained over time.

There is still a lot of work to be performed on the benefits and restrictions that apply to coaching. This must be done without forgetting the psychological premises that help us to understand it. Indeed, when understood as a construct, an underlying scientific foundation must be developed.

In addition, longitudinal studies will be necessary for future research, at least throughout the academic year, and would allow us to obtain information that closely monitors the reality of the changes that occur not only in the sample under study, but also in the control group. Likewise, it will be pertinent to test whether the results obtained in the school coaching process improve or are maintained according to certain particularities, such as academic performance, the size of the group, and the temporality of the sessions in particular and of the coaching process in general.

Finally, it will be advisable to take a deeper look at the scientific models on which coaching is based for a closer examination of this methodology.

## 6. Conclusions

On the basis of this study, it can be concluded that there was a non-significant increase in the global scale with which the effective personality may be assessed, mainly under the dimensions of strengths, demands, and relationships with oneself for the experimental group. In terms of gender, there was a tendency for improvement in the dimensions of challenges among males and a slight tendency for improvement in strengths among females, which was perhaps due to the greater degree of autonomy given to adolescent boys compared to girls [44,71]. On the other hand, the course year/age variable was not associated with different perceptions of the functioning of the program.

In relation to the type of educational center (state-aided/state schools), a tendency can be seen among the respondents that pointed to better functioning of the program in the area of relationships. Finally, in the qualitative analysis, the students’ perception of this intervention was favorable. The students were able to reflect on important situations through this activity that had not been previously performed. They performed working on their self-conception and interpersonal relationships. Focusing on strengths, demands, and relationships, all the measures implemented favored a state of emotional wellbeing [21,68,69].

The teachers likewise mentioned that they were able to perceive an important improvement in the interpersonal relationships between students and student-and-tutors. This positively improved the classroom atmosphere and favored by the increase of intrapersonal variables such as student self-esteem and academic self-conception [85].

## Figures and Tables

**Figure 1 children-10-01025-f001:**
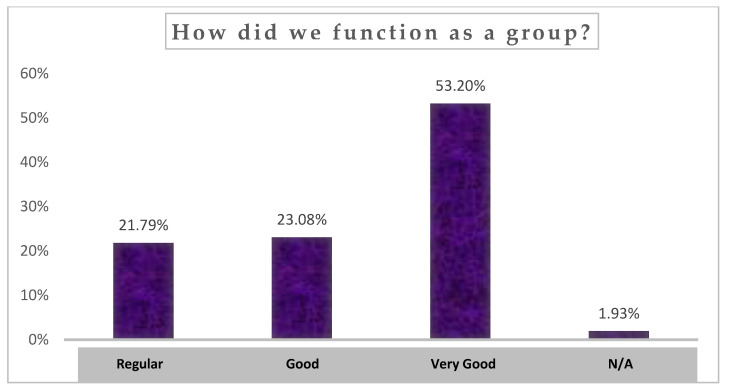
Student perceptions of how the group functioned during the school group coaching intervention. Note: N/A: no answer. (Source: Authors’ own work).

**Figure 2 children-10-01025-f002:**
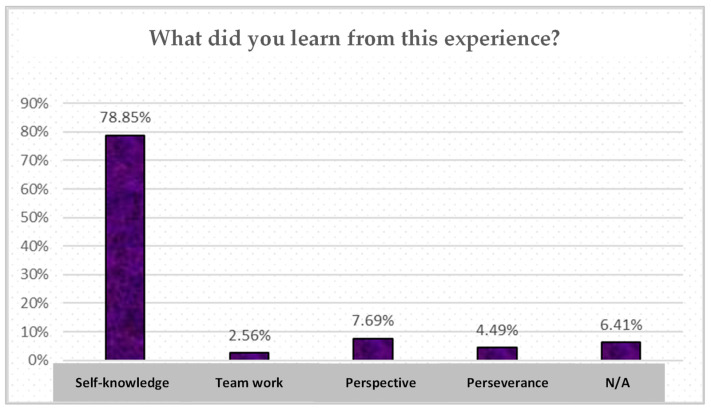
Student perceptions of learning during the school group coaching intervention. Note: N/A: No answer. (Source: Authors’ own work).

**Figure 3 children-10-01025-f003:**
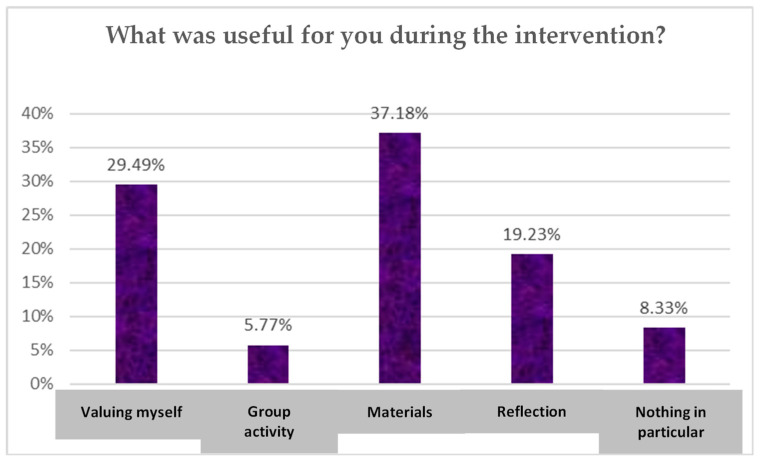
Students’ perceptions of the benefits of the school group coaching intervention. (Source: Authors’ own work).

**Figure 4 children-10-01025-f004:**
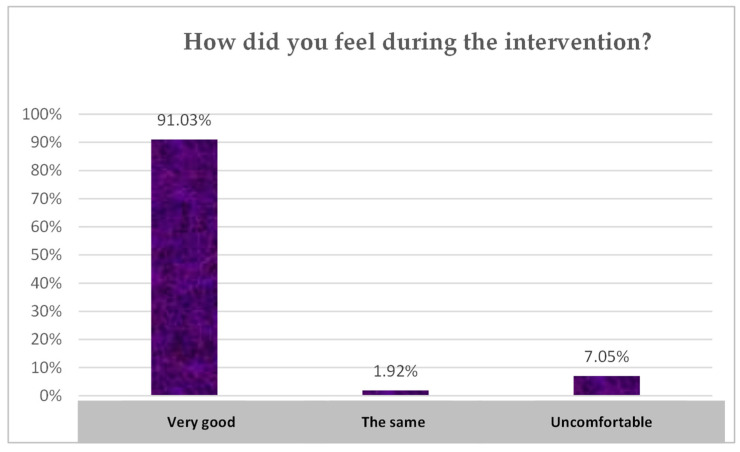
Approximation of the students’ feelings during the school group coaching interventions. (Source: Authors’ own work).

**Figure 5 children-10-01025-f005:**
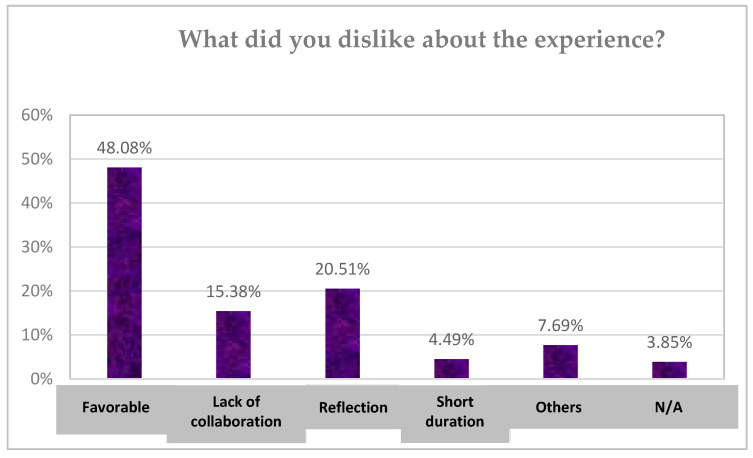
Perceived levels of student satisfaction during the school group coaching intervention. Note: N/A: No answer. (Source: Authors’ own work).

**Figure 6 children-10-01025-f006:**
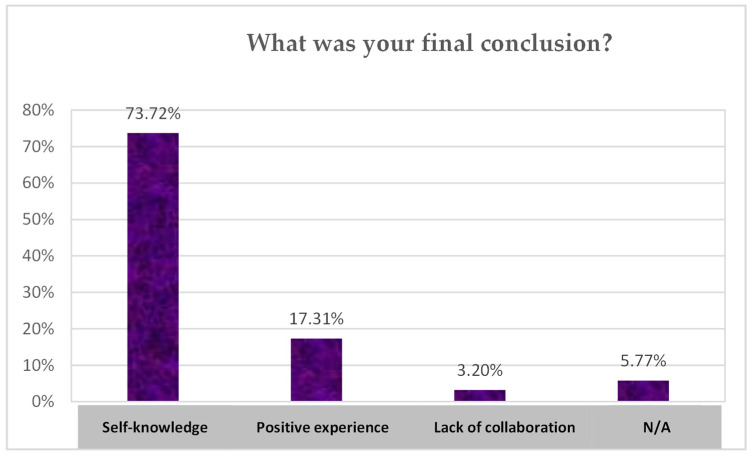
Final conclusions of the students on the school group coaching intervention. Note: N/A: No answer. (Source: Authors’ own work).

**Table 1 children-10-01025-t001:** Phases of the School Coaching Process.

Session Number	Purpose of Activity	Duration of Intervention
Session 1	Inform participants of the process and establish empathy and commitment.	45–50 min
Session 2	Foster relationships through active listening and group participation.	45–50 min
Session 3	Increase trust, respect, and listening within the group.	45–50 min
Session 4	Manage emotions and introspection through reflection to generate new paradigms and models of action.	45–50 min
Session 5	Develop an action plan to incorporate new behaviors.	45–50 min
Session 6	Motivation, feedback, and remedying what the learner has learned. General conclusion.	45–50 min

Source: Authors’ own work.

**Table 2 children-10-01025-t002:** Descriptive statistics of the control and the experimental group in the Pre-test and the Post-test.

	Control		Experimental	
	Average	td.	Average	td.
Effective Personality Pre-test	84.6234	12.87247	85.5897	10.52976
Effective Personality Post-test	83.7078	12.49676	85.8269	10.34824
Strengths Pre-test	15.2338	3.31523	15.4423	2.72205
Strengths Post-test	15.1883	3.23940	15.6538	2.74662
Demands Pre-test	28.3571	5.89414	28.7179	5.69640
Demands Post-test	28.0065	5.88284	28.7564	5.86776
Challenges Pre-test	17.8442	3.98384	18.1218	3.44981
Challenges Post-test	17.4740	3.71923	17.9936	3.52410
Relationships Pre-test	23.1883	4.69288	23.3077	3.87484
Relationships Post-test	23.0390	4.49238	23.4231	3.70882

**Table 3 children-10-01025-t003:** ANCOVA analysis of the effectiveness of the school coaching intervention, in accordance with the global scale and dimensions of effective personality.

		F	gl.	*p*	*η* ^2^ * _p_ *
Global scale and areas	Effective Personality	2.717	1	0.100 ^+^	0.009
	Strengths	1.730	1	0.189	0.006
	Demands	1.286	1	0.258	0.004
	Challenges	1.193	1	0.276	0.004
	Relationships	0.909	1	0.341	0.003

+ Marginal significance (at 0.100 level).

**Table 4 children-10-01025-t004:** ANCOVA test results of the effectiveness of the school group coaching intervention, according to the variables gender, course year/age, and type of educational center (state-aided/state schools).

	Gender				Course Year/Age				Type of Educational Centre			
	F	gl.	*p*	*η* ^2^ * _p_ *	F	gl.	*p*	*η* ^2^ * _p_ *	F	gl.	*p*	*η* ^2^ * _p_ *
Effective Personality	1.354	1	0.246	0.009	0.478	2	0.621	0.006	0.566	1	0.453	0.004
Strengths	2.954	1	0.088 ^+^	0.019	0.015	2	0.985	0.000	1.769	1	0.185	0.011
Demands	0.009	1	0.925	0.000	0.113	2	0.893	0.001	0.013	1	0.909	0.000
Challenges	4.660	1	0.032 *	0.030	0.419	2	0.659	0.005	0.399	1	0.529	0.003
Relationships	0.094	1	0.759	0.001	0.241	2	0.786	0.003	3.247	1	0.074 ^+^	0.021

* Significance at a level of 0.05. ^+^ Marginal significance (at a level of 0.100).

## Data Availability

Not available.

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
