# Peer review of "Impact of Coaching on the Development of Personal and Social Competences among Secondary School Students"

_children, 2023, doi:10.3390/children10061025_

Round 1

Reviewer 1 Report

Dear Sir/Mam

Please find bellow the requested review regarding the manuscript. The article contains a lot of useful information on the issue. The topic is very interesting and but use of sources is not appropriate. Although it has some useful information there are less references and the statements are not established. I suggest the authors to write more information with references.

The article contains a lot of useful information on the issue. It is quite clear what is already known about this topic and the research question is clearly outlined. The abstract is too brief and introduction section involves too many information. The research question is not justified clearly, given what is already known about the topic. The results are not discussed from multiple angles and conclusions answer the aims of the study partially. The conclusions are partially supported by references or results and the limitations of the study fatal and it is questionable if there are opportunities to inform future research. Positive: There are some strengths of the article that could have an impact in the field, such as the topic and its impact on the existed literature. 

Dear Sir/Mam

Please find bellow the requested review regarding the manuscript. The article contains a lot of useful information on the issue. The topic is very interesting and but use of sources is not appropriate. Although it has some useful information there are less references and the statements are not established. I suggest the authors to write more information with references.

The article contains a lot of useful information on the issue. It is quite clear what is already known about this topic and the research question is clearly outlined. The abstract is too brief and introduction section involves too many information. The research question is not justified clearly, given what is already known about the topic. The results are not discussed from multiple angles and conclusions answer the aims of the study partially. The conclusions are partially supported by references or results and the limitations of the study fatal and it is questionable if there are opportunities to inform future research. Positive: There are some strengths of the article that could have an impact in the field, such as the topic and its impact on the existed literature. The manuscript is approved publication only after major changes.

Author Response

Dear Sir/Mam

Please find below the requested review regarding the manuscript. The article contains a lot of useful information on the issue. The topic is very interesting and but use of sources is not appropriate. Although it has some useful information there are less references and the statements are not established. I suggest the authors to write more information with references.

  • Thank you for your comment, the appropriate correction has been made. It is worth mentioning that the length of the abstract is within the parameters commented in the template: Abstract: A single paragraph of about 200 words maximum.
  • Thank you for your comments, this has been taken into account and the introduction has been reworded with specific information.
  • Thank you for your comments: We have proceeded to include and modify information with more references that you will find throughout the document.
  • Please take note: Table 1: Phases of the coaching process is replaced by Table 1: Phases of the School Coaching Process as it was included in error, please note the latter.

The article contains a lot of useful information on the issue. It is quite clear what is already known about this topic and the research question is clearly outlined. The abstract is too brief and the introduction section involves too many information. The research question is not justified clearly, given what is already known about the topic. The results are not discussed from multiple angles and conclusions answer the aims of the study partially. The conclusions are partially supported by references or results and the limitations of the study fatal and it is questionable if there are opportunities to inform future research. Positive: There are some strengths of the article that could have an impact in the field, such as the topic and its impact on the existed literature. 

  • Thank you for your comment, the appropriate correction has been made. It is worth mentioning that the length of the abstract is within the parameters commented in the template: Abstract: A single paragraph of about 200 words máximum.
  • The research question is clarified and justified by the new information that has been included in the paper.
  • Both the findings, discussion and conclusions have been expanded with current research on the topic.
  • The heading of limitations and future implications is included to give greater visibility to this.

Comments on the Quality of English Language

Dear Sir/Mam

Please find below the requested review regarding the manuscript. The article contains a lot of useful information on the issue. The topic is very interesting and but use of sources is not appropriate. Although it has some useful information there are less references and the statements are not established. I suggest the authors to write more information with references.

 The article contains a lot of useful information on the issue. It is quite clear what is already known about this topic and the research question is clearly outlined. The abstract is too brief and introduction section involves too many information. The research question is not justified clearly, given what is already known about the topic. The results are not discussed from multiple angles and conclusions answer the aims of the study partially. The conclusions are partially supported by references or results and the limitations of the study fatal and it is questionable if there are opportunities to inform future research. Positive: There are some strengths of the article that could have an impact in the field, such as the topic and its impact on the existed literature. The manuscript is approved publication only after major changes.

  • Thank you for your comment, the appropriate correction has been made. It is worth mentioning that the length of the abstract is within the parameters commented in the template: Abstract: A single paragraph of about 200 words máximum.
  • Thank you for your comments, this has been taken into account and the introduction has been reworded with specific information.
  • Thank you for your comments: We have proceeded to include and modify information with more references that you will find throughout the document.
  • Please take note: Table 1: Phases of the coaching process is replaced by Table 1: Phases of the School Coaching Process as it was included in error, please note the latter.

Reviewer 2 Report

A longitudinal design was carried out by evaluating the secondary school students to find the impact of coaching on the development of their personal and social competences.
It is not necessary to use the references in the abstract.
Line 8... correct "de, velopment".
Lines 54-60... This paragraph appears to be written in smaller characters. Reference "pp,, 35" should also be corrected.
Line 111...   You should not put pages next to the references "[31, pp. 852]".
Line 141... Delete the point.
Line 155... Use capital letter.
Line 200... The title of figure 1 is not complete.
Line 208... The title of figure 2 is not complete.
Line 222... The title of figure 4 is not complete.
Line 229... The title of figure 5 is not complete.
Line 236... The title of figure 6 is not complete.
Please review the work from the technical editing point of view. I also recommend that you ask for the help of an authorized translator or a native speaker.
The references are appropriate, the article presents 76 references, being up to date. Some years are written in bold, others are not. Please write them all the same.

Author Response

A longitudinal design was carried out by evaluating the secondary school students to find the impact of coaching on the development of their personal and social competences.It is not necessary to use the references in the abstract.

  • Thank you for your comment, the appropriate correction has been made
  • All these suggestions have been taken into account and you will find them rectified in each case. Thank you for your suggestions to improve the content of this article.

Line 8... correct "de, velopment".   
Lines 54-60... This paragraph appears to be written in smaller characters. Reference "pp,, 35" should also be corrected. 
Line 111...   You should not put pages next to the references "[31, pp. 852]".  
Line 141... Delete the point.  
Line 155... Use capital letter. 
Line 200... The title of figure 1 is not complete.  
Line 208... The title of figure 2 is not complete.   
Line 222... The title of figure 4 is not complete.    
Line 229... The title of figure 5 is not complete.    
Line 236... The title of figure 6 is not complete.    
Please review the work from the technical editing point of view. I also recommend that you ask for the help of an authorized translator or a native speaker.
The references are appropriate, the article presents 76 references, being up to date. Some years are written in bold, others are not. Please write them all the same.  

corrected.

Round 2

Reviewer 2 Report

The authors have corrected/completed all recommendations.